Citation: *Molecular Systems Biology* 9:670
www.molecularsystemsbiology.com

# REPORT

# Synthetic mammalian transgene negative autoregulation

**Vinay Shimoga[1,2,5], Jacob T White[1,2,5], Yi Li[1,2], Eduardo Sontag[3] and Leonidas Bleris[1,2,4,*]**

[1] Bioengineering Department, The University of Texas at Dallas, Richardson, TX, USA, [2] Center for Systems Biology, The University of Texas at Dallas, Richardson, TX, USA, [3] Department of Mathematics, Rutgers University, New Brunswick, NJ, USA and [4] Electrical Engineering Department, The University of Texas at Dallas, Richardson, TX, USA
[5]These authors contributed equally to this work
* Corresponding author. Electrical Engineering, Bioengineering Department, The University of Texas at Dallas, 800 West Campbell Road, NSERL 4.708, Richardson, TX 75080, USA. Tel.: +1 972 883 5785; Fax: +1 972 883 5785; E-mail: bleris@utdallas.edu

**Biological networks contain overrepresented small-scale topologies, typically called motifs. A frequently appearing motif is the transcriptional negative-feedback loop, where a gene product represses its own transcription. Here, using synthetic circuits stably integrated in human kidney cells, we study the effect of negative-feedback regulation on cell-wide (extrinsic) and gene-specific (intrinsic) sources of uncertainty. We develop a theoretical approach to extract the two noise components from experiments and show that negative feedback results in significant total noise reduction by reducing extrinsic noise while marginally increasing intrinsic noise. We compare the results to simple negative regulation, where a constitutively transcribed transcription factor represses a reporter protein. We observe that the control architecture also reduces the extrinsic noise but results in substantially higher intrinsic fluctuations. We conclude that negative feedback is the most efficient way to mitigate the effects of extrinsic fluctuations by a sole regulatory wiring.**
*Molecular Systems Biology* **9**: 670; published online 4 June 2013; doi:10.1038/msb.2013.27
*Subject Categories:* metabolic and regulatory networks; synthetic biology
*Keywords:* cellular noise; human cells; negative feedback; transgenes

## Introduction

Information in cells propagates through intricate, diverse biochemical pathways. Within these complex networks certain small-scale interaction patterns appear more frequently than others (Milo *et al*, 2002; Alon, 2006, 2007). The convergence of pathways to particular motifs may be attributed to their inherent topological and functional properties, with a range of theoretical and experimental results supporting this hypothesis.

A network motif of particular interest is negative feedback, which appears in high frequency in bacterial (Alon, 2006), yeast (Lee *et al*, 2007), and mammalian cells (Odom *et al*, 2006). The negative-feedback loop consists of a single node that represses its own synthesis, and has been shown to accelerate transcriptional response time (Rosenfeld *et al*, 2002) and reduce gene expression noise in bacteria and yeast (Thattai and van Oudenaarden, 2001; Becskei and Serrano, 2002; Dublanche *et al*, 2006; Nevozhay *et al*, 2009). Remarkably, theoretical and experimental results show that negative feedback might either amplify or reduce noise in gene expression (Thattai and van Oudenaarden, 2001; Simpson *et al*, 2003; Austin *et al*, 2006; Cox *et al*, 2008; Nacher and

Ochiai, 2008; Singh and Hespanha, 2009; Marquez-Lago and Stelling, 2010), highlighting the need for additional experimental investigation, particularly in human cells.

Even though endogenous motifs are composed of relatively few elements, they are typically embedded as 'modules' in larger networks that exhibit complex behavior. Therefore, synthetic gene circuits, orthogonal to endogenous cellular processes, are a suitable experimental platform for elucidating their topological and functional properties.

Events controlling synthesis and degradation are independent for different proteins in a cell, and are often called 'intrinsic' or 'local' noise (Elowitz *et al*, 2002; Blake *et al*, 2003; Paulsson, 2004; Raser and O'Shea, 2005). A strongly expressing constitutive promoter is expected to have little intrinsic noise, while a weak promoter will have high intrinsic noise (Bar-Even *et al*, 2006; Newman *et al*, 2006). These variations propagate along pathways, with the consequence that protein distributions along a pathway appear correlated (Swain *et al*, 2002; Volfson *et al*, 2006). However, even proteins from different regulation pathways may show correlation, owing to stochastic variations in quantities that affect the regulation of all genes (Swain *et al*, 2002; Volfson *et al*, 2006), such as polymerase copies. As a consequence, two identical, independently

regulated promoters are expected to have the same extrinsic noise, which originates through global effects (Elowitz *et al*, 2002; Swain *et al*, 2002; Raser and O'Shea, 2005).

A two-reporter experimental platform (Elowitz *et al*, 2002) has been instrumental for studying extrinsic and intrinsic noise as well as pathway-specific effects (Colman-Lerner *et al*, 2005; Pedraza and van Oudenaarden, 2005). Results in *Escherichia coli* and yeast cells show that extrinsic noise dominates the total noise (Colman-Lerner *et al*, 2005), especially at high protein abundances (Taniguchi *et al*, 2010), whereas fluctuations at low-protein copy numbers are owing to intrinsic noise. A limitation of this experimental platform is the requirement for two identically regulated reporters with equal variances. To overcome this constraint, we develop an approach that permits intrinsic and extrinsic noise breakdown for two non-identical reporters, and use this methodology to study noise in synthetic mammalian transgene negative autoregulation.

## Results

### Integration of the circuits and initial characterization

We integrated the negative autoregulation and the control architectures (Figure 1) in Tet-On immortalized human kidney cells (Materials and methods, Generation of stable cell lines). We originally engineered (Bleris *et al*, 2011) these architectures using a bidirectional promoter that transcribes two genes fused in multiple cloning sites (MCSI and MCSII) upstream from minimal CMV promoters. The bidirectional promoter is activated in the presence of Doxycycline (Dox) by the transcription factor rtTA that, for all reported experiments, is produced stably from the cell line.

For the implementation of the control architecture (Figure 1A), we chose the transcriptional repressor LacI to be cloned in the MCSI and we fused sequences containing a tandem repeat of the wild-type LacO between the Pcmv region and the start codon of the dsRed monomer reporter gene (fused in the MCSII). We fused the zsGreen1 protein upstream from LacI, using an internal ribosome entry site (IRES), a nucleotide sequence that allows for translation initiation in the middle of a messenger RNA (mRNA) sequence. We constructed the transcriptional negative autoregulatory motif by inverting the promoter region of the control motif copying the wild-type LacO sequence (Figure 1B). As a result, the LacI protein inhibits the transcription of its own mRNA (and the co-expressed zsGreen1), and the constitutive output is now measured by the dsRed fluorescent. The negative-feedback strength can be tuned by IPTG induction.

Subsequently, we integrated the circuits stably in cells (Materials and methods, Generation of stable cell lines). For fixed Dox in a monoclonal cell population, the architecture depicted in Figure 1A is a simple negative regulation and serves as the control. We note that by changing the Dox levels, the output of the control architecture (Figure 1A) will depend on both X (rtTA) and Y (LacI) thereby emulating a Type I incoherent feedforward architecture (Alon, 2006). Previously (Bleris *et al*, 2011), we transiently transfected plasmids carrying these circuits and studied the behavior of the reporter proteins. Our experiments showed that the output node of an incoherent feedforward motif is largely invariant to the changes in the DNA fragment (i.e., primarily the copy number).

We first induced the stable clones with a wide range of IPTG and Dox concentrations, and the output was quantified after 24 h using flow cytometry (Materials and methods, Data processing). A gate based on the forward and side scatter is

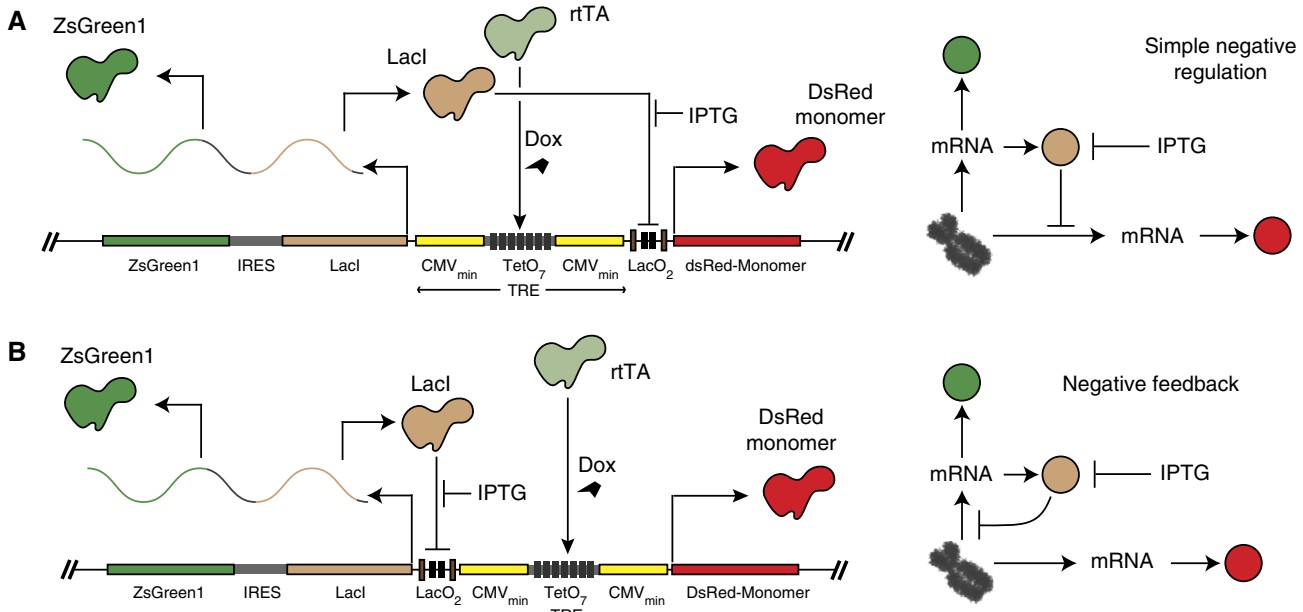

**Figure 1** The synthetic architectures integrated in human kidney cells. (**A**) The control architecture: the bidirectional promoter under the control of rtTA transcribes the ZsGreen1–IRES–LacI and dsRed monomer transcripts. The dsRed monomer is inhibited by LacI. (**B**) The negative feedback: The bidirectional promoter under the control of rtTA transcribes the ZsGreen1–IRES–LacI and dsRed monomer transcripts. The ZsGreen1–IRES–LacI transcript is inhibited by LacI.

first used to select single cell events followed by a gate that retains the constitutive protein-positive events (i.e., dsRed for the negative feedback and zsGreen1 for the control architecture) at the threshold of negative cells (Supplementary Figure 1).

A Dox titration of the negative-feedback architecture shows that the mean protein concentration increases for both the constitutive side, dsRed, and the side controlled by negative feedback. In the case of dsRed, we have a three-fold increase, while for zsGreen1 we have a two-fold increase in the mean protein concentration (Supplementary Figure 2a). For the control architecture, the mean protein concentration increased by five-fold the constitutive side (zsGreen1) and 2.5-fold the regulated side, dsRed (Supplementary Figure 2b). These results are in agreement with our transient transfection experiments (Bleris *et al*, 2011), and show that the incoherent feedforward architecture is superior to negative feedback in controlling the output mean levels due to changes in the input.

We then performed IPTG titrations (0.0625–25 μM) at two Dox concentrations, 625 ng/ml (defined as 'low') and 5000 ng/ml (defined as 'high'). The Dox values were selected in order to study the effect of the expression level on the total noise, and to provide sufficient separation of the mean output levels (Supplementary Figure 3). Importantly, the ability to control the transcription levels of the transgene allows us to emulate the effect of the variable transcriptional activity

expected at different genomic locations. Consistently with our experiments (Supplementary Figure 3), previous experimental (Bar-Even *et al*, 2006) and theoretical (Paulsson, 2004) studies show that noise scales inversely with the protein abundance.

For the negative feedback, the titrations performed at low and high concentrations of Dox show a corresponding five-fold increase for the high Dox case and a three-fold increase in the low Dox case in mean fluorescence of zsGreen1 (Figure 2A). As expected, the dsRed protein levels remains constant over the entire range of IPTG concentrations (Figure 2B). We provide selected microscopy snapshots of the induced negative-feedback populations in Figure 2C. The IPTG titrations of the simple negative regulation clone result to a four-fold and eight-fold increase in dsRed protein levels at low and high Dox concentrations, respectively, whereas the constitutively synthesized protein levels were unchanged (Figure 2D and E, respectively). We provide selected microscopy snapshots of the induced simple regulation populations in Figure 2F. We also include all the flow cytometry data and the corresponding histograms for the IPTG titrations for both transgenes and two Dox conditions (Supplementary Figures 4–7).

To further probe the behavior of the circuits, we quantified the number of copies of integrations for our circuits using real-time quantitative PCR (Materials and methods, and Supplementary Material, Integration Copies). We found that

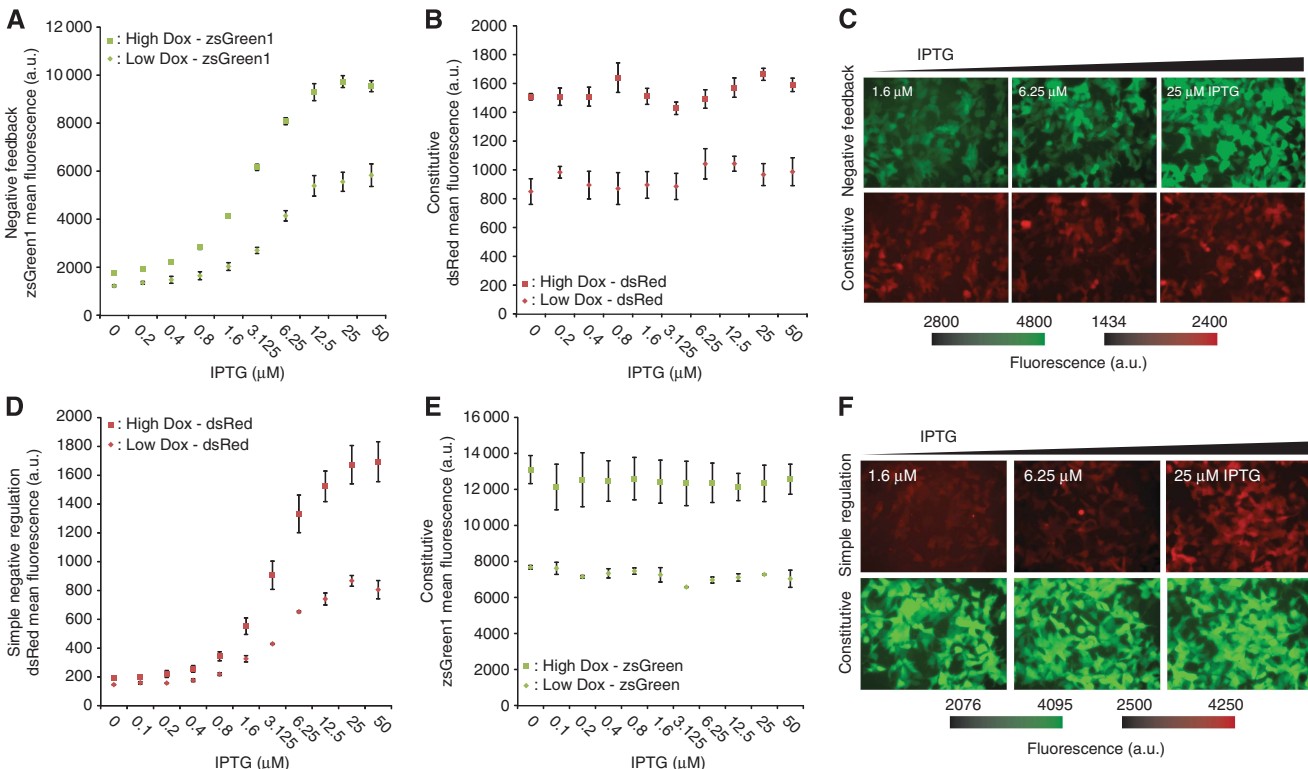

**Figure 2** IPTG titrations for the negative feedback and simple regulation transgenes. (**A–C**) The negative feedback: zsGreen protein in green (under negative autoregulation) and dsRed in red (constitutively synthesized). Saturated Dox concentration with squares and low DOX concentration with diamonds. Error bars show the s.d. of triplicate experiments. (**A**) The mean zsGreen fluorescence. (**B**) The mean dsRed fluorescence. (**C**) Microscopy images of IPTG titrations at high concentration of Dox. (**D–F**) Simple regulation architecture: the dsRed in red (under regulation) and the zsGreen protein in green (constitutively synthesized); saturated Dox concentration with squares and low DOX concentration with diamonds. Error bars show the s.d. of triplicate experiments. (**D**) The mean dsRed fluorescence. (**E**) The mean zsGreen fluorescence. (**F**) Microscopy images of IPTG titrations at high concentration of Dox. Source data for this figure is available on the online supplementary information page.

the negative-feedback clone is a single integration while the simple negative regulation is two copies (Supplementary Material, Table I). We created a new transgene of the simple negative regulation with a single integration and showed that their behavior is consistent (Supplementary Figure 8).

## Analysis of noise

After the initial characterization of the transgenes, our next objective was to extract the two noise components from the experimental data. The total noise observed in a fluorescent reporter distribution arises through the combination of global (extrinsic) fluctuations together with the fluctuations in that protein's local regulation machinery (intrinsic). In the standard two-reporter formulation (Elowitz *et al*, 2002), the extrinsic noise becomes the normalized covariance of the two reporters that are independently regulated and identically distributed. We extend this analysis to cases where it is not feasible to construct two identically regulated reporters or it is impossible to obtain identical reporter statistics.

We use a multiplicative noise model (Supplementary Material, Theory), where the total fluctuations of one reporter are the product of an extrinsic random variable and intrinsic random variable, while the second reporter fluctuations are the product of the same extrinsic random variable but its own intrinsic random variable. The three-model components are assumed independent. As described in detail in the Supplementary Material, we convert this multiplicative model to a linear model and show that the extrinsic noise is the normalized covariance of two constitutive reporters and the intrinsic noise is the difference between the observed CV square and the extrinsic CV square. Importantly, to decouple the extrinsic noise of the regulated and constitutive reporters, we add the sensitivity coefficient α to the extrinsic random variable for the regulated reporter.

We define as Y the constitutive reporter, X the regulated reporter (controlled by an inducer, in our case IPTG), and α is the coefficient that is 1 for two constitutive promoters with identical reporter statistics but varies depending on the regulation of X. Considering the case of negative feedback, when LacI is fully inactivated by IPTG at saturation, both reporters are equally sensitive to extrinsic fluctuations; thereby the extrinsic noise will be the normalized covariance as defined previously (Elowitz *et al*, 2002). We can safely postulate that the unregulated reporter should have equal sensitivity to extrinsic fluctuations for all IPTG conditions. Consequently, we calculate the regulated reporter sensitivity coefficient α for each sample in the IPTG titration such that the extrinsic noise of the unregulated reporter is the same as in the full IPTG well (Supplementary Material, Table II). In summary, we use the following noise breakdown:

$$n_{\text{extX}}^2 = \alpha_i \text{Cov}(X_i, Y_i) \tag{1}$$

$$n_{\text{intX}}^2 = n_{\text{totX}}^2 - n_{\text{extX}}^2 \tag{2}$$

$$n_{\text{extY}}^2 = \text{Cov}(X_i, Y_i)/\alpha_i \tag{3}$$

$$n_{\text{intY}}^2 = n_{\text{totY}}^2 - n_{\text{extY}}^2 \tag{4}$$

$$\alpha_N = \frac{n_{\text{totX}}}{n_{\text{totY}}} \tag{5}$$

$$\alpha_i = \frac{\text{Cov}(X_i, Y_i)\alpha_N}{\text{Cov}(X_N, Y_N)}, i \neq N \tag{6}$$

where $n_{\text{totX}}^2$ and $n_{\text{totY}}^2$ are experimentally determined CV squares of reporters X and Y, $\text{Cov}(X_i, Y_i)$ is the covariance of the logarithms of X and Y, and the inducer concentrations are indexed $i = 1, 2, \ldots, N$, where well $N$ is fully induced.

## The effect of negative-feedback regulation on noise

We next obtain the intrinsic, extrinsic, and total noise of the two architectures (Figure 1) for the IPTG titrations of Figure 2. The negative-feedback architecture experiments show that stronger feedback (i.e., low IPTG concentration) reduces extrinsic noise (and the total noise) but mildly increases intrinsic noise (Figure 3A and D for high and low Dox, respectively). As expected, the noise levels remain flat for the control protein output (Figure 3C and F), with the extrinsic noise significantly higher than the intrinsic noise. In addition, for the negative-feedback architecture, when comparing the high and low Dox cases (Figure 3A and D), we observe that the strong induction leads to lower total noise. We also plot the noise versus the mean levels of the output protein, to probe directly the impact of negative feedback. Indeed, as illustrated in Figures 3B and E the negative feedback reduces noise. In contrast, the simple negative regulation remains in the same range for decreasing mean (Figure 3H).

When we examine the noise breakdown of the simple negative regulation architecture, we observe that the noise scales with protein abundance (Figure 3G for high Dox and Supplementary Figure 9 for low Dox). The effect is not as pronounced as with the negative feedback but we partially attribute this to a post-processing of the data, performed in order to discard the portion of the events that merge with the background signal (Supplementary Figures 6–7). The post-processing is particularly necessary for the low Dox case and although qualitatively consistent these results are not taken into consideration (Supplementary Figures 7 and 9). We emphasize that for the negative autoregulation there is no post-processing beyond the constitutive protein gating (Supplementary Figures 4–5).

For the simple negative regulation architecture, the results show that stronger regulation (i.e., low IPTG concentration) reduces extrinsic noise but significantly, as compared with the negative feedback, increases intrinsic noise (Figure 3G), resulting to approximately flat total noise. As expected, the control constitutive protein noise breakdown remains flat for high Dox (Figure 3I).

In order to validate our noise decomposition, we performed alternative analyses of the raw experimental data and simulations. First, we used a method to filter out extrinsic noise, simply by processing the flow cytometry data using a smaller forward versus side scatter gate (Newman *et al*, 2006). This approach has been reported to filter out extrinsic noise in

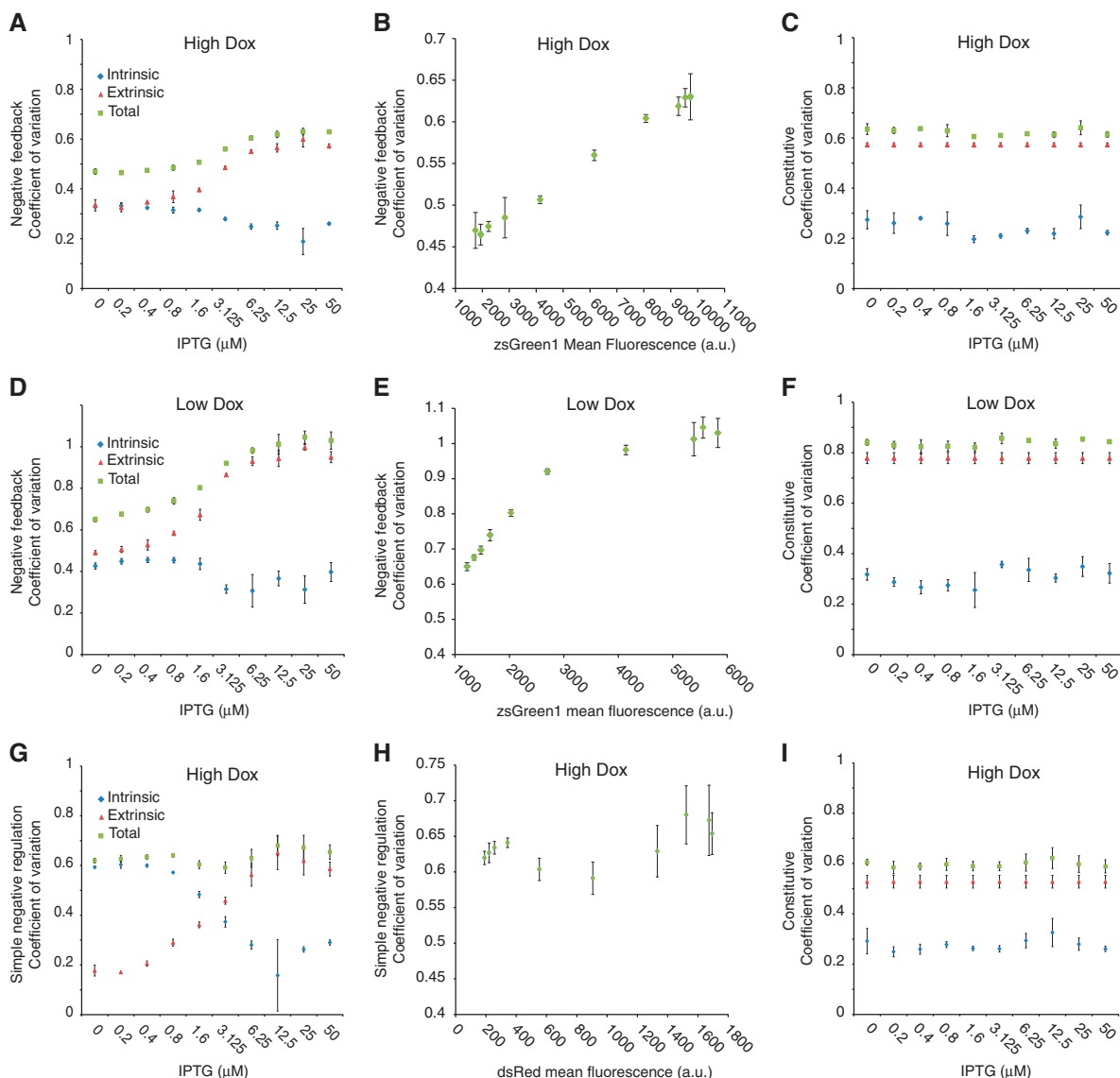

**Figure 3** Noise for the negative-feedback loop and the simple regulation transgenes. Intrinsic noise with blue diamonds, extrinsic noise with red triangles, and total noise with green squares. Error bars are the s.d. of triplicates. (**A–F**) Negative-feedback architecture: (**A**) coefficient of variation for high Dox. (**B**) Coefficient of variation versus mean zsGreen1 protein levels for high Dox. (**C**) Coefficient of variation for the constitutive dsRed protein for high Dox. (**D**) Coefficient of variation for low Dox. (**E**) Coefficient of variation versus mean zsGreen1 protein levels for low Dox. (**F**) Coefficient of variation for the constitutive dsRed protein for low Dox. (**G–I**) Simple regulation architecture: (**G**) coefficient of variation for the dsRed protein for high Dox. (**H**) Coefficient of variation versus mean dsRed protein levels for high Dox. (**I**) Coefficient of variation for the constitutive zsGreen1 protein for high Dox. Source data for this figure is available on the online supplementary information page.

mammalian cells (Singh *et al*, 2010). Indeed, we find (Supplementary Figure 10) that reducing the gate decreased the total noise due to a decrease in the extrinsic noise, while the intrinsic noise remains the same. Furthermore, the overall trend of the drop in noise with decreasing concentrations of IPTG did not change. We also used simulations to gain additional insight into the ways our method is able to decompose noise. Specifically, as discussed in the Supplementary Material Theory section, we first vary the strength of transcription of a single bidirectional promoter coding for two fluorescent proteins, leading to perfectly correlated fluorescence quantities, which our decomposition shows to have only extrinsic noise and no intrinsic noise. Next, we vary the transcriptional activity of two fluorescent genes

independently, which leads to uncorrelated fluorescence quantities; our method returns only intrinsic noise and no extrinsic noise. Subsequently, we show that the decomposition is correct for mixtures of intrinsic and extrinsic noise.

Finally, in order to further validate the consistency of our results, we examined the relationship between the chromosomal position of our transgene and the experimentally observed phenotype. We created new integrations for the negative feedback, selected three random colonies, and we performed titrations of IPTG for high Dox levels. A direct comparison between the main and new clones (Supplementary Figure 11) shows that the overall behavior is conserved with the absolute noise levels being marginally different. This difference is expected considering that distant

chromosomal sites often have significant differences in their transcriptional activity (Dar *et al*, 2012).

# Discussion

Investigating the relationship between regulatory systems and cellular noise has inherently wide biological significance. Our results shed new light on one of the most abundant biological motifs, the negative-feedback loop. In particular, we used synthetic circuits stably integrated in human kidney cells and we studied the effect of negative feedback on cell-wide and gene-specific sources of uncertainty.

We developed an approach to extract the extrinsic and intrinsic noise contributions from experimental measurements of two reporter proteins that are controlled by non-identical promoters. Our experiments reveal that negative feedback reduces extrinsic noise while slightly increasing intrinsic noise. Importantly, negative feedback reduces the total noise. By comparing these results to simple negative regulation, we argue that negative feedback is the most efficient way to reduce extrinsic fluctuations by introducing a sole additional regulatory wiring.

It has been shown theoretically (Simpson *et al*, 2003) and confirmed experimentally (Austin *et al*, 2006) that negative autoregulation can filter out lower frequency noise. In addition, it has been shown experimentally (Rosenfeld *et al*, 2005) that extrinsic fluctuations have lower frequency components than intrinsic noise. It follows that negative autoregulation would remove extrinsic noise. Our results indeed show that negative autoregulation removes extrinsic noise; however, we also observe that simple negative regulation also removes extrinsic noise, de-correlating the negatively regulated reporter from the constitutive reporter. Time lapse experiments can be used to shed additional light to the properties of these architectures.

To conclude, our analysis shows that negative feedback reduces noise, but only from extrinsic sources, outside of the genetic components of the feedback loop. Furthermore, we show that the negative feedback raises intrinsic noise but the cost of this regulation is small when compared with simple negative regulation.

# Materials and methods

## Generation of stable cell lines

Tet-ON cells (Clontech), which stably express the transcription factor rtTA, were used for all the experiments. Cells were grown in 12-well plates (Greiner Bio-One) at 80% confluency and transfected (LTX Transfection reagent, Invitrogen) with the plasmid carrying the circuit and a Hygromycin linear selection marker (Clontech) in a 1:20 ratio. Cells were transferred to Petri dishes and incubated in 25 µg/ml hygromycin for a week. Hygromycin was then reduced to 15 µg/ml and cells were incubated until colonies developed. Colonies that were positive for florescent signals were selected by microscopy and picked using cloning rings and further expanded.

## Cell culture

The cells were grown at 37°C and 5% $CO_2$. The cells were grown in Dulbecco's modified Eagle's medium (DMEM, Invitrogen, Cat # 11965-11810) supplemented with 0.1 mM MEM non-essential amino acids

(Invitrogen, Cat # 11140-050), 0.045 units/ml of penicillin and 0.045 µg/ml streptomycin (Penicillin–Streptomycin liquid, Invitrogen), and 10% fetal bovine serum (FBS, Invitrogen). The adherent culture was maintained in this medium by trypsinizing with Trypsin–EDTA (0.25% Trypsin with EDTAx4Na, Invitrogen) and diluting in a fresh medium upon reaching 50–90% confluence.

## Flow cytometry

The cells were prepared for flow cytometry by trypsinizing each well with 0.5 ml 0.25% trypsin–EDTA, collecting the cell suspension, and centrifuging at 4000 r.p.m. for 2 min. Trypsin was removed and the pellet resuspended by short vortexing in 0.5 ml PBS buffer (Invitrogen). Cells were run on a LSR Fortessa (BD Biosciences) Flow cytometer equipped with the FACSDiva software program. One hundred thousand cells were counted in each run. DsRed was measured with a 561 nm laser and a 586 nm emission filter with a 582/15 band pass filter, and ZsGreen1 with a 488 nm laser and a 509 nm emission filter with a 515/20 band pass filter. Data analysis was performed using FlowJo and Matlab.

## Data processing

First, we use the constitutive fluorescence protein-positive population of cells, based on a control experiment in which cells are uninduced (Supplementary Figure 1). Second, to remove outliers (explained in the supplement), we include cells that have a fluorescent intensity equal to 2.5 times the s.d. of the population.

## Microscopy

All microscope images were taken from live cells grown in multi-well plates (Greiner Bio-One) in the DMEM supplemented with non-essential amino acids, penicillin/streptomycin, and 10% FBS. Cells were imaged using the Olympus IX81 microscope and a Precision Control environmental chamber. The images were captured using a Hamamatsu ORCA–03 Cooled monochrome digital camera. The filter sets (Chroma) are as follows: ET470/50x (excitation) and ET525/50m (emission) for ZsGreen1, ET560/40x (excitation) and ET630/75m (emission) for DsRed. For the negative feedback, the exposure times were ZsGreen1:500 ms and dsRed:1000 ms, while for the simple negative regulation, the exposure times were ZsGreen1:400 ms and dsRed:2000 ms. Data collection and processing was performed in the software package Slidebook 5.0 and Adobe Illustrator.

## Copy number using real-time PCR

We performed real-time quantitative PCR to determine the absolute copies of integration for our circuits. The average copy numbers of dsRed of all stable clones were estimated by the delta delta Ct method as follows: $2^{-\Delta\Delta Ct} = ((1 + E_{DsRED})^{-\Delta Ct,DsRED}) / ((1 + E_{BRCA1})^{-\Delta Ct, BRCA1})$, where $E_{DsRED}$ is the PCR amplification efficiency for dsRed and $E_{BRCA1}$ for BRCA1 (endogenous reference gene)(Zheng *et al*, 2011). A control stable HEK293 cell line was generated by Flp-In system (Invitrogen) and contains a single copy of dsRed transgene (Li *et al*, 2012). To determine the PCR amplification efficiency, genomic DNAs from the control cell line were used to generate the dilution curve of $\log_2$(DNA amount, ng) versus Ct. $E_{DsRED}$ was calculated as 1.07, and $E_{BRCA1}$ as 0.98. For each stable clone, triplicates (50 ng of genomic DNA) were performed and the average copy numbers were calculated as the mean ± s.d.

## Supplementary information

# Acknowledgements

This work was supported by a US National Institutes of Health (NIH) grant 1R15GM096271, a Texas Analog Center of Excellence (TxACE) grant P12095, a National Science Foundation (NSF) award (CBET-1105524), and the University of Texas at Dallas. We thank the anonymous reviewers for their constructive comments and feedback.

*Author contributions:* VS performed the integrations and measurement experiments. YL performed the copy-number quantification experiments. JTW, LB, ES, and VS developed the theory. VS, JTW, LB, and YL analyzed the data. JTW, LB, VS, ES, and YL prepared the manuscript. LB designed the experiments and supervised the project.

# Conflict of interest

The authors declare that they have no conflict of interest.

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
