## [Review Process File · Molecular Systems Biology]

Synthetic mammalian transgene negative autoregulation

Vinay Shimoga, Jacob T. White, Yi Li, Eduardo Sontag and Leonidas Bleris

Corresponding author: Leonidas Bleris, University of Texas at Dallas

Review timeline:	Submission date:	15 October 2012
	Editorial Decision:	30 November 2012
	Revision received:	26 February 2013
	Editorial Decision:	17 April 2013
	Revision received:	02 May 2013
	Accepted:	03 May 2013

Editor: Thomas Lemberger

Transaction Report:

1st Editorial Decision

30 November 2012

Thank you again for submitting your work to Molecular Systems Biology. We have now heard back from the three referees who agreed to evaluate your manuscript. As you will see from the reports below, the referees find the topic of your study of potential interest. They raise, however, substantial concerns on your work, which, I am afraid to say, preclude its publication in its present form.

The reviewers thus raise the following major concerns that should be convincingly addressed with additional experimentation and analyses:

- A major concern with regard to the experimental design refers to the use of stable integrant clones. Since effects due to variable integration sites preclude a rigorous comparison between circuits across different clones, this point should be convincingly addressed.
- Given the methodological focus of the study, your approach should be compared to alternative methods.
- Simulation data should be analysis to gain more precise insights in to the ways your method is able to decompose noise.

On a more editorial level, we would also ask you to reformat the manuscript to the short Report format (3 figures maximum <http://www.nature.com/msb/authors>).

Given the quantitative nature of the work, we would also ask you to provide the quantitative experimental data that are represented in the figure of your papers and used for the analysis. These data should ideally be provided as 'source data'

REFeree REPORTS:

Reviewer #1 (Remarks to the Author):

The authors address the effects of negative autoregulation on stochastic noise in mammalian cells. The authors devise a new, elegant, experimental approach to analyze extrinsic versus intrinsic noise in negative feedback and develop the analytic theory to support this approach. The analysis in mammalian cells is particularly noteworthy and impressive and the associated theory to extract extrinsic and intrinsic noise components for any gene circuit could be a powerful tool.

While I liked the approach, I was not overly excited/surprised by the results. There are two theoretical papers that demonstrate the possible increase and decrease of noise by negative autoregulation that were overlooked and should be cited in the introduction: Simpson et al., PNAS (2003) and Cox et al., PNAS (2008). In addition, the modulation of extrinsic noise by negative autoregulation was demonstrated by Austin et al., Nature (2006). There the ATc mediated shift was extrinsic noise related and negative feedback shifted the frequency range of the total noise.

Therefore, I think the strength of the paper is primarily on the approach and the paper needs to be rewritten to emphasize the experimental approach over the findings. For a journal of MSB's caliber, this entails a more rigorous experimental analysis to compare this approach to other existing experimental approaches. In short, I would be supportive of publication after a major revision (and some sort of benchmarking against others methods). This will require a change in title to focus on the method over the results. Below, I address additional technical issues and minor issues that should be addressed in the revised manuscript.

Major technical issues:

1. The authors must address the effect of integration site on noise. A major study recently showed that noise varies considerably at different integration sites across the human genome (Dar et al., PNAS, 2012) and the authors make no mention of this integration site behavior. They are testing constructs that have integrated into different integration sites (an inherent outcome of colony selection after hygromycin selection), which may have significantly different intrinsic noise as a result of transcriptional bursting. I appreciate that the high IPTG level measurements and their non-feedback control but the observations could easily be particular to the specific integration sites selected. Can one really compare the different circuits at different sites? The authors really must address this issue.

2. The approach should be benchmarked against at least one other approach. One possible avenue that would be quite easy to carry out (since it is simply reanalysis of the existing flow cytometry data) is to use a previous method to filter out extrinsic noise by taking a very small gate in forward scattering vs. side scattering (Newman et al., Nature, 2006). This approach has been reported to filter out extrinsic noise in mammalian cells (Singh et al. Biophys J, 2010). Total noise is the large gate while intrinsic noise is the small gate. I would like to know how the results in Fig. 4 compare between the two approaches. At the very least, this would strengthen the paper since the authors could comment on the shortcomings of this older method.

3. Negative feedback is a high-pass filter and extrinsic noise is lower frequency than intrinsic noise, so it would be the first to be filtered out. This is, in part, what I mean the result not being too surprising. The authors make no mention of this concept addressed in: Simpson et al., PNAS (2003) and Cox et al., PNAS (2008), Austin et al., Nature (2006). This should be discussed.

Minor issues:

1. The theory section "The theoretical methodology to extract..." is written very much like a method rather than a Results section. This section appears to have been "tacked on" (Fig. 4b is cited before Fig. 4a) and is not well integrated with the other sections, which is disconcerting for the reader. I suggest this section be significantly shortened to focus on the most relevant concepts and results and that the bulk of the derivations be moved to the methods or an appendix/SI section so that this section reads less like a methods section.

2. I did not understand the reason for the micrographs in Figs 2c and 3c. They are too qualitative and not helpful to understanding. These should be replaced with quantitative flow cytometry or microscopy histograms showing both the percentage of the population responding and the

expression level in the responding population.

Reviewer #2 (Remarks to the Author):

The authors present a theoretical framework for extracting the "intrinsic" and "extrinsic" contributions to gene expression noise, and apply their formalism to the analysis of a synthetic gene circuit built in cultured human cells. The subject matter and methods used are very appropriate for the readership of MSB. However, I found the manuscript deficient in a number of ways. These deficiencies need to be addressed before publication can be considered.

My main concern is that there is no feedback between the theoretical and experimental components; they appear as two separate endeavors. The theoretical formalism is developed, and then applied to the experimental data, but there is no "closing of the loop": no theoretical predictions that are then tested experimentally, no comparison between theory and experiment, etc. As a result, it is hard to assess the validity of the theory.

If direct comparison between the theory and experimental data is too challenging, then the authors should use numerical simulation on which to test their formalism. Such a simulation, using standard stochastic models of gene activity to describe their synthetic circuit, will be less phenomenological than their theory while still being simpler than the experimental data. It may thus allow them to test their theoretical tools and provide intuition for the results later obtained from real experimental data.

Specific comments:

1. "The negative feedback [...] reduce gene expression noise in bacteria and yeast" (page 3). Against this point of view, there is a significant body of evidence (in both *E. coli* and yeast) suggesting that the observed level of gene expression noise is merely a function of the mean expression level, while being independent of the circuit topology (including the presence/absence of feedback). See e.g. Bar Even et al., *Nature Genetics* 2006 (cited by the authors); So et al., *Nature Genetics* 2011; Salman et al., *PRL* 2012; Hensel et al., *NSMB* 2012.

Later in the manuscript, when the authors analyze the effect of feedback on gene expression noise (page 10 and Figure 4), the noise is never plotted against the mean level. This makes it impossible to assess the effect of presence/absence of feedback. Plotting against IPTG level is not very insightful. A plot of noise versus mean for the different cases needs to be added. If, after examination, the authors still believe that the IPTG plot is more helpful, than the noise/mean plot can be placed in the SI.

2. "A strongly expressing constitutive promoter is expected to have little intrinsic noise, while a weak promoter will have high intrinsic noise [...] protein distributions along a pathway appear correlated [...] proteins from different regulation pathways may show correlation, due to stochastic variations in quantities that affect the regulation of all genes, such as polymerase copies" (page 3). Each of these statements requires a citation in support.

3. When introducing the synthetic gene circuit (page 4), it was unclear to me whether one or more copies of the circuit were integrated into the genome. What is the copy number? Perhaps this was hidden somewhere in the Methods, but it should be stated in the main text.

4. The two paragraphs describing the authors' previous usage of the gene circuit (page 5, beginning with "Previously, we transiently transfected...") were very confusing. What exactly is the relevance of this discussion to the current work? Some sentences were incomprehensible, for example: "... the gene output of the circuit will exhibit a particular function with respect to the DNA fragment".

5. In developing their theoretical formalism, the authors state that "the total fluctuations of one reporter are the product of an extrinsic random variable and intrinsic random variable, while the second reporter fluctuations are the product of the same extrinsic random variable but its own intrinsic random variable" (page 6). No evidence (or references) is presented to support this statement. In particular, how is this simple phenomenology justified based on the standard stochastic models of gene expression kinetics?

6. Figures 1-3. A scale bar is missing from the fluorescence images.

Reviewer #3 (Remarks to the Author):

The work by Shimoga et al explores how negative feedback reduces noise in a mammalian genetic circuit. They infer that negative feedback reduces extrinsic noise but slightly increases intrinsic noise leading to a reduction of the total noise. To reach this conclusion, they use an altered two-color assay in which not only the reporters are different but even the promoters. One of the reporter is regulated by a repressor embedded in a negative feedback while the other by a constitutive promoter. To make this comparison possible they show a theoretical analysis of covariance assuming a multiplicative noise with parameters comparable to logarithmic sensitivity. I had encountered the following difficulties to assess this work:

A. The separation of noise into intrinsic and extrinsic noise has no fixed physical and biological meaning and assumptions of the opposite has led to difficulties to compare different works with different set ups. In the classical two-reporter studies by Elowitz et al, most extrinsic noise comes from the repressor. As Paulsson (2005, Physics of life reviews) points out "if the two reporters were regulated by different repressors, repressor fluctuations would move from the extrinsic to the intrinsic category". Since in the circuits constructed by Shimoga et al, only one of the two reporters is regulated by the repressor, the above change of categories applies.

B. Secondly, it is not clear how good is the approximation used in the current approach since no validation is provided. A number of alternatives to the presented noise decomposition can be imagined.

1. In order to clarify the meaning and validity of the authors' noise decomposition, they should perform stochastic simulations (e.g. Gillespie's SSA). For this, it is recommended using assumptions on specific cellular processes and parameters that reproduce the experimental results in order to show systematically how noise is reconstructed from the simulated observations. The decomposition of simulated noise can be then compared to the approach presented by the authors. This will be also helpful to avoid difficulties with the categorization of intrinsic and extrinsic noise so that comparison to future works with different genetic constructions and noise decomposition becomes possible.

Further comments:

2. The authors do not discuss possible molecular mechanisms behind the decomposed intrinsic and extrinsic noise. Do the authors aim at pure phenomenological noise decomposition (correlated and uncorrelated noise)?

3. Some of the terminology is confusing. Why are the synthesis and degradation of proteins named thermal events (supplementary information)? Which processes in a cell are thermal events and which are not?

4. The authors do not specify if the constructs are chromosomally integrated. If yes, they should verify where the integration events occurred. It would be useful to show results from multiple different chromosomal integrations. This would certainly help to illustrate the decomposition of noise associated with the different chromosomal localizations (see. Eg. Becskei et al (2005, Nature Genetics), Batenchuk et al (2011, Biophysical J)).

5. The authors should provide the value of background fluorescence of cells not containing any GFP and RFP in Fig. 1.

6. It is quite unusual that the total noise reduces with increasing mean expression (Fig. 4a, c). Do the authors have an explanation for this?

7. More details should be provided for the experimental noise decomposition. Each step (15-20) should be presented in the supplementary information. For example, what the value for alpha is.

8. It would be very helpful to plot the noise terms against the mean expression level and not only against IPTG.

Point-by-point response to reviewers' comments

We appreciate all the comments and remarks and thank the reviewers for their in-depths analysis of our manuscript. We believe that we have been able to address all reviewers' concerns, as detailed in the enclosed point-by-point response and as reflected in the revised version of this manuscript.

Our additional analysis corroborated our initial assertions, and the inclusion of the new data has indeed strengthened our conclusions. We hope that the reviewers will find our answers satisfactory. Please note that in the enclosed document the original comments are in blue color.

For all the co-authors,
Leonidas Bleris

Reviewer #1 (Remarks to the Author):

The authors address the effects of negative autoregulation on stochastic noise in mammalian cells. The authors devise a new, elegant, experimental approach to analyze extrinsic versus intrinsic noise in negative feedback and develop the analytic theory to support this approach. The analysis in mammalian cells is particularly noteworthy and impressive and the associated theory to extract extrinsic and intrinsic noise components for any gene circuit could be a powerful tool.

While I liked the approach, I was not overly excited/surprised by the results. There are two theoretical papers that demonstrate the possible increase and decrease of noise by negative autoregulation that were overlooked and should be cited in the introduction: Simpson et al., PNAS (2003) and Cox et al., PNAS (2008). In addition, the modulation of extrinsic noise by negative autoregulation was demonstrated by Austin et al., Nature (2006). There the ATc mediated shift was extrinsic noise related and negative feedback shifted the frequency range of the total noise.

Therefore, I think the strength of the paper is primarily on the approach and the paper needs to be rewritten to emphasize the experimental approach over the findings. For a journal of MSB's caliber, this entails a more rigorous experimental analysis to compare this approach to other existing experimental approaches. In short, I would be supportive of publication after a major revision (and some sort of benchmarking against others methods). This will require a change in title to focus on the method over the results. Below, I address additional technical issues and minor issues that should be addressed in the revised manuscript.

We thank the reviewer for his remarks and analysis. We strived to address all his comments. We proceeded with adding the three references that discuss theoretical results in the manuscript. Furthermore, we revised significantly the paper, placing the theory in the supplement. The title of the manuscript is now “Synthetic Mammalian Transgene Negative Autoregulation”, shifting the focus from the noise analysis to the methods associated with the characterization of the transgene. To address the specific technical issues and other comments we used additional experiments and theoretical analysis.

Major technical issues:

1. The authors must address the effect of integration site on noise. A major study recently showed that noise varies considerably at different integration sites across the human genome (Dar et al., PNAS, 2012) and the authors make no mention of this integration site behavior. They are testing constructs that have integrated into different integration sites (an inherent outcome of colony selection after hygromycin selection), which may have significantly different intrinsic noise as a result of transcriptional bursting. I appreciate that the high IPTG level measurements and their non-feedback control but the observations could easily be particular to the specific integration sites selected. Can one really compare the different circuits at different sites? The authors really must address this issue.

The relationship between the chromosomal position of a transgene and the noise is a subject of great interest to us and the community. This is a question which we actively pursue under the grant that is supporting this work. Even with several clones at different locations it is non-trivial without significant number of additional experiments to answer with high confidence the general question if one can compare circuits at different sites.

To partially answer the question, we believe that indeed one can qualitatively compare circuits at different locations. We subscribe to the notion that architectures that are abundant in nature must carry inherent topological properties that are conserved independently of the location in the genome. Clearly distant chromosomal sites can have different “activity” (which may be transcriptional activity or epigenetic modifications). To support our argument we performed experiments to check for the consistency in behavior of the circuits at different locations. It is conceivable that the absolute levels of noise will be different for distant genomic locations, but if the behavior is conserved then one can (at least) qualitatively compare circuits at different sites.

To probe the behavior across clones for the negative feedback architecture we created new integrations and we performed titrations of IPTG for high doxycycline levels. As illustrated in the following figure, a direct comparison between three clones shows that the behavior between the transgenes is indeed consistent. Specifically, negative feedback reduces noise (**Figure 1**).

Moreover, a new transgene (Clones II) has slightly weaker expression levels, yet the noise levels in response to IPTG are similar. In contrast Clone III has identical expression profile with Clone I yet the total noise is higher. The constitutive side has the expected flat mean protein levels (versus IPTG) and similar levels of noise. Again the Clone III has higher levels of noise which can be attributed to the location, as also corroborated by the calculation of the intrinsic noise

(Figure 2). We also built an additional clone of the simple regulation circuit. Again, the overall behavior remains consistent.

Figure 1: Comparison of mean fluorescence and coefficient of variation between the main paper negative feedback clone and two new transgens.

■ Negative feedback Clone I (main paper) ▲ Negative feedback Clone II ◆ Negative feedback Clone III

Figure 2: Comparison of intrinsic and extrinsic breakdown for the main paper negative feedback clone and two new transgenes.

Additionally, in the first version of the manuscript we included the following experiment (**Figure 3**) in the supplement, but we now emphasize its importance and relevance to the question. The figure corresponds to the coefficient of variation for different strength of transcription as controlled by the levels of doxycycline for the main paper negative feedback clone. We argue that our ability to control the transcription levels of the transgene emulates to some extent the effect of variable transcriptional activity of the transgene. Again, the behavior is consistent.

Figure 3. Coefficient of variation versus IPTG for the negative feedback loop for various concentrations of doxycycline.

To summarize, using a set of transgenes, we show that the architectures have consistent behavior in different locations of the genome. The general question warrants further investigation but based on these results we argue that the qualitative behavior of the transgenes is conserved. We added this discussion in the main text and reference to Supplement Figure 11.

2. The approach should be benchmarked against at least one other approach. One possible avenue that would be quite easy to carry out (since it is simply reanalysis of the existing flow cytometry data) is to use a previous method to filter out extrinsic noise by taking a very small gate in forward scattering vs. side scattering (Newman et al., Nature, 2006). This approach has been reported to filter out extrinsic noise in mammalian cells (Singh et al. Biophys J, 2010). Total noise is the large gate while intrinsic noise is the small gate. I would like to know how the results in Fig. 4 compare between the two approaches. At the very least, this would strengthen the paper since the authors could comment on the shortcomings of this older method.

Indeed we used the method suggested by the reviewer to filter out extrinsic noise by taking a very small gate in forward scattering vs. side scattering (Newman et al., Nature, 2006). We found that reducing the gate decreased the total noise due to a decrease in the extrinsic noise. The intrinsic noise remains the same. Furthermore, the overall trend of the drop in noise with decreasing concentrations of IPTG did not change.

Here will provide the two gates for comparison side by side as requested (**Figure 4**). Panels a and b correspond to the original gate (SSC 20k-120k and FSC 20k-120k gate, Supplement Figure 1) while panels c and d were prepared using smaller gate (SSC 40k-50k and FSC 70k-80k gate).

Figure 4: Effect of forward scattering vs. side scatter gate on the extrinsic noise.

We thank the reviewer for the comment and we added the following paragraph in the main paper: “In order to validate our noise decomposition we performed an alternative analysis of the raw experimental data and simulations. Specifically, we used a method to filter out extrinsic noise, simply by processing the flow cytometry data using a smaller forward versus side scatter gate¹. This approach has been reported to filter out extrinsic noise in mammalian cells². Indeed we find (**Supplementary Fig. 10**) that reducing the gate decreased the total noise due to a

decrease in the extrinsic noise and the intrinsic noise remains the same. Furthermore, the overall trend of the drop in noise with decreasing concentrations of IPTG did not change.”

3. Negative feedback is a high-pass filter and extrinsic noise is lower frequency than intrinsic noise, so it would be the first to be filtered out. This is, in part, what I mean the result not being too surprising. The authors make no mention of this concept addressed in: Simpson et al., PNAS (2003) and Cox et al., PNAS (2008), Austin et al., Nature (2006). This should be discussed.

As the reviewer points out, it has been shown theoretically³ and confirmed experimentally⁴ that negative autoregulation can filter out lower frequency noise. Additionally, it has been shown experimentally⁵ that extrinsic fluctuations have lower frequency components than intrinsic noise. It follows that negative autoregulation would remove extrinsic noise. Our results indeed show that negative autoregulation removes extrinsic noise; however, we also observe that simple negative regulation also removes extrinsic noise, decorrelating the negatively regulated reporter from the constitutive reporter. We thank the reviewer for the remark. We added the above discussion to the manuscript (Remarks section). In the future, we plan to use time-lapse experiments to shed additional light to the properties of these architectures.

Minor issues:

1. The theory section "The theoretical methodology to extract..." is written very much like a method rather than a Results section. This section appears to have been "tacked on" (Fig. 4b is cited before Fig. 4a) and is not well integrated with the other sections, which is disconcerting for the reader. I suggest this section be significantly shortened to focus on the most relevant concepts and results and that the bulk of the derivations be moved to the methods or an appendix/SI section so that this section reads less like a methods section.

We revised significantly the paper, placing the theory in the supplement.

2. I did not understand the reason for the micrographs in Figs 2c and 3c. They are too qualitative and not helpful to understanding. These should be replaced with quantitative flow cytometry or microscopy histograms showing both the percentage of the population responding and the expression level in the responding population.

We removed the microscopy snapshots from Figure 1. We agree that the microscopy snapshots are qualitative but we believe that they provide an important dimension to the manuscript, a direct connection to what is studied.

We include supplement images of the flow cytometry data for all architectures (Supplement Figures 4-7). For example the following figure for the negative feedback under high doxycycline conditions.

Figure 5. IPTG Titrations for the negative feedback loop, under high Doxycycline. DsRed positive cells are illustrated and the corresponding histograms for each output are also presented. ■

To address the concern of the reviewer and minimize the space used, yet include these images we selected a subset that illustrates (qualitatively) the effect of the IPTG addition for both circuits (main paper Figure 2 and 3).

We emphasize here that all the data presented in the main paper Figures 2 and 3 are flow cytometry data processed as described in the Methods and Supplementary material sections.

Reviewer #2 (Remarks to the Author):

Review: Shimoga et al.

The authors present a theoretical framework for extracting the "intrinsic" and "extrinsic" contributions to gene expression noise, and apply their formalism to the analysis of a synthetic gene circuit built in cultured human cells. The subject matter and methods used are very appropriate for the readership of MSB. However, I found the manuscript deficient in a number of ways. These deficiencies need to be addressed before publication can be considered.

My main concern is that there is no feedback between the theoretical and experimental components; they appear as two separate endeavors. The theoretical formalism is developed, and then applied to the experimental data, but there is no "closing of the loop": no theoretical predictions that are then tested experimentally, no comparison between theory and experiment, etc. As a result, it is hard to assess the validity of the theory.

If direct comparison between the theory and experimental data is too challenging, then the authors should use numerical simulation on which to test their formalism. Such a simulation, using standard stochastic models of gene activity to describe their synthetic circuit, will be less phenomenological than their theory while still being simpler than the experimental data. It may thus allow them to test their theoretical tools and provide intuition for the results later obtained from real experimental data.

We thank the reviewer for his insightful and constructive comments.

Specific comments:

1. "The negative feedback [...] reduce gene expression noise in bacteria and yeast" (page 3). Against this point of view, there is a significant body of evidence (in both *E. coli* and yeast) suggesting that the observed level of gene expression noise is merely a function of the mean expression level, while being independent of the circuit topology (including the presence/absence of feedback). See e.g. Bar Even et al., *Nature Genetics* 2006 (cited by the authors); So et al., *Nature Genetics* 2011; Salman et al., *PRL* 2012; Hensel et al., *NSMB* 2012. Later in the manuscript, when the authors analyze the effect of feedback on gene expression noise (page 10 and Figure 4), the noise is never plotted against the mean level. This makes it impossible to assess the effect of presence/absence of feedback. Plotting against IPTG level is not very insightful. A plot of noise versus mean for the different cases needs to be added. If, after examination, the authors still believe that the IPTG plot is more helpful, than the noise/mean plot can be placed in the SI.

We thank the reviewer for his comment. Indeed we plotted the noise against the mean levels and included the following panels in the main paper Figure 3. We believe that both plots (noise/mean and noise/IPTG) are informative, thereby we kept both.

Figure 6: The coefficient of variation versus the mean expression levels of the negative feedback.

2. "A strongly expressing constitutive promoter is expected to have little intrinsic noise, while a weak promoter will have high intrinsic noise [...] protein distributions along a pathway appear correlated [...] proteins from different regulation pathways may show correlation, due to stochastic variations in quantities that affect the regulation of all genes, such as polymerase copies" (page 3). Each of these statements requires a citation in support.

We thank the reviewer for the remark. We updated the paragraph with new references and discussion: “A strongly expressing constitutive promoter is expected to have little intrinsic noise, while a weak promoter will have high intrinsic noise^{1,6}. These variations propagate along pathways, with the consequence that protein distributions along a pathway appear correlated^{7,8}. However, even proteins from different regulation pathways may show correlation, due to stochastic variations in quantities that affect the regulation of all genes^{7,8}, such as polymerase copies.”

3. When introducing the synthetic gene circuit (page 4), it was unclear to me whether one or more copies of the circuit were integrated into the genome. What is the copy number? Perhaps this was hidden somewhere in the Methods, but it should be stated in the main text.

We performed real-time quantitative PCR to determine the copies of the circuits. Various studies have demonstrated that this method is of satisfactory accuracy compared to Southern blot⁹. For example, Table 2 adapted from “Determination of Cytochrome P450 2D6 (CYP2D6) gene copy number by real-time quantitative PCR”¹⁰, the estimations of CYP2D6 gene copies from real-time quantitative PCR match with those from Southern blotting.

The average copy numbers of DsRED of all stable clones were estimated by the delta delta Ct method as follows: $2^{-\Delta\Delta Ct} = ((1 + E_{DsRED})^{-\Delta Ct, DsRED}) / ((1 + E_{BRCA1})^{-\Delta Ct, BRCA1})$, where E_{DsRED} is the PCR amplification efficiency for DsRED and E_{BRCA1} for BRCA1 (endogenous reference gene)¹¹. The control stable HEK293 cell line was generated by Flp-In system (Invitrogen) and contains one copy of DsRED transgene¹². To determine the PCR amplification efficiency, genomic DNAs from the control cell line were used to generate the dilution curve of $\log_2(\text{DNA amount, ng})$ vs. Ct. E_{DsRED} was calculated as 1.07, and E_{BRCA1} as 0.98. For each stable clone, triplicates (50 ng of

genomic DNA) were performed and the average copy numbers were calculated as the mean \pm SD.

Clone	Gene Copy Average	Standard Deviation
Negative Feedback	1.026716891	0.205356713
Simple Negative Regulation	1.986616106	0.063272704
Single integration clone	1.005016457	0.127722277

We included these results in the main text and additional discussion in the supplement.

To further examine the effect of the two copies of integration for our simple regulation architecture we created new transgenes and present here the comparison with a transgene that has a single integration. In this case, the level of expression for the 2 integrations is higher than a single integration. Similar to the main paper clone low Doxycycline experiments the dsRed signal for the single clone merges with the background signal for strong repression. Therefore, considering the IPTG concentrations that we are above the background signal we show that the coefficient of variation for the two clones is consistent. The results were included as relevant discussion in the main text and Supplementary Figure 8 (included the raw flow cytometry data).

Figure 7: Comparison of mean fluorescence and coefficient of variation between the main paper simple regulation clone and a new transgene.

4. The two paragraphs describing the authors' previous usage of the gene circuit (page 5, beginning with "Previously, we transiently transfected...") were very confusing. What exactly is the relevance of this discussion to the current work? Some sentences were incomprehensible, for example: "... the gene output of the circuit will exhibit a particular function with respect to the DNA fragment".

We restructured these paragraphs to the following: "Subsequently, we integrated the circuits stably in cells (Methods, Generation of stable lines). For fixed doxycycline in a monoclonal cell population, the architecture depicted in Figure 1a is a simple negative regulation and serves as the control. We note that by changing the doxycycline levels the output of the control architecture (Fig. 1a) will depend on both X (rtTA) and Y (LacI) thereby emulating a Type I incoherent feedforward architecture¹³. Previously¹⁴, we transiently transfected plasmids carrying these circuits and studied the behavior of the reporter proteins. Our experiments showed that the output node of an incoherent feedforward motif is largely invariant to the changes in the DNA fragment (i.e. primarily the copy number)."

5. In developing their theoretical formalism, the authors state that "the total fluctuations of one reporter are the product of an extrinsic random variable and intrinsic random variable, while the second reporter fluctuations are the product of the same extrinsic random variable but its own intrinsic random variable" (page 6). No evidence (or references) is presented to support this statement. In particular, how is this simple phenomenology justified based on the standard stochastic models of gene expression kinetics?

We motivate the selection of the multiplicative model as follows. Suppose gene X is activated by two factors; one (A) is an intrinsic variable such as a transcription factor, and the other (B) is an extrinsic variable, such as RNA polymerase. Suppose both factors must be present for transcription, in the complex ABX. We have four reaction equations:

This results in the following differential equations at steady-state:

$$\begin{aligned} k_{r1}[AX] + k_{r2}[BX] &= k_{f1}[A][X] + k_{f2}[B][X] \\ k_{f1}[A][X] + k_{r3}[ABX] &= k_{r1}[AX] + k_{f3}[B][AX] \\ k_{f2}[B][X] + k_{r4}[ABX] &= k_{r2}[BX] + k_{f4}[A][BX] \\ k_{f4}[A][BX] + k_{f3}[B][AX] &= k_{r4}[ABX] + k_{r3}[ABX] \end{aligned}$$

For the gene activity, we take the ratio of active complex ABX to total gene copies:

$$X_{active} = \frac{[ABX]}{[X] + [AX] + [BX] + [ABX]}$$

This simplifies to an expression in terms of A and B (we drop most of the constants):

$$X_{active} = \frac{[A][B] + [A]^2[B] + [A][B]^2}{k + [A] + [B] + [A]^2 + [B]^2 + [A]^2[B] + [A][B]^2}$$

Which, for small, unsaturated concentrations of A and B, looks like:

$$X_{active} \approx \frac{[A][B]}{k}$$

Intuitively, in this multiplicative approximation, a polymerase fluctuation of 10% is expected to change gene activity by 10% (with an unsaturated promoter). Compare this to an additive noise model: now, the same polymerase fluctuation of 1000 molecules is expected to change gene output by 1000 molecules, regardless of whether the output is currently regulated at 10000 molecules or at 100 molecules. Thus the multiplicative model makes physical sense for positive variables, where a reporter with 100 molecules cannot have an uncertainty of 1000 molecules.

We included the above analysis in the Supplementary Material Theory section.

6. Figures 1-3. A scale bar is missing from the fluorescence images.

We added scale bars to the fluorescent images. Furthermore in the Methods section we included the following: “For the negative feedback, the exposure times were ZsGreen1:500ms and dsRed:1000ms, while for the simple regulation, the exposure times were ZsGreen1:400ms and dsRed:2000ms.”

Reviewer #3 (Remarks to the Author):

The work by Shimoga et al explores how negative feedback reduces noise in a mammalian genetic circuit. They infer that negative feedback reduces extrinsic noise but slightly increases intrinsic noise leading to a reduction of the total noise. To reach this conclusion, they use an altered two-color assay in which not only the reporters are different but even the promoters. One of the reporter is regulated by a repressor embedded in a negative feedback while the other by a constitutive promoter. To make this comparison possible they show a theoretical analysis of covariance assuming a multiplicative noise with parameters comparable to logarithmic sensitivity. I had encountered the following difficulties to assess this work:

A. The separation of noise into intrinsic and extrinsic noise has no fixed physical and biological meaning and assumptions of the opposite has led to difficulties to compare different works with different set ups. In the classical two-reporter studies by Elowitz et al, most extrinsic noise comes from the repressor. As Paulsson (2005, Physics of life reviews) points out "if the two reporters were regulated by different repressors, repressor fluctuations would move from the extrinsic to the intrinsic category". Since in the circuits constructed by Shimoga et al, only one of the two reporters is regulated by the repressor, the above change of categories applies.

B. Secondly, it is not clear how good is the approximation used in the current approach since no validation is provided. A number of alternatives to the presented noise decomposition can be imagined.

We thank the reviewer for his insightful and constructive comments. We agree with the reviewer assessment and understand the limitations and shortcomings of the extrinsic/intrinsic formulations. As a result we avoided introducing an extended discussion about the molecular mechanisms behind the noise contributions. Even in the main paper discussion, we deliberately omit interpreting the results and classifying potential noise contributions. We proceed with addressing his comments.

1. In order to clarify the meaning and validity of the authors' noise decomposition, they should perform stochastic simulations (e.g. Gillespie's SSA). For this, it is recommended using assumptions on specific cellular processes and parameters that reproduce the experimental results in order to show systematically how noise is reconstructed from the simulated observations. The decomposition of simulated noise can be then compares to the approach presented by the authors. This will be also helpful to avoid difficulties with the categorization of intrinsic and extrinsic noise so that comparison to future works with different genetic constructions and noise decomposition becomes possible.

In our noise decomposition, we expect random quantities which affect the expression of both genes to show up as extrinsic noise, while we expect random quantities which affect only a

single gene to show up as intrinsic. We address the case where one reporter may be less sensitive to extrinsic noise sources due to noise-reducing regulatory pathways. To see this, first take the simplest case of a two-color experiment: suppose we have a plasmid with a constitutive bidirectional promoter P coding for reporters X and Y, and let the only source of uncertainty be the plasmid copy number (or transcriptional strength). Then we have production rates of each reporter:

$$\begin{aligned}\frac{dX}{dt} &= k_1P - k_2X \\ \frac{dY}{dt} &= k_3P - k_4Y\end{aligned}$$

At steady-state, we have the relations

$$\begin{aligned}X &= \frac{k_1}{k_2}P \rightarrow \log(X) = \log\left(\frac{k_1}{k_2}\right) + \log(P) \\ Y &= \frac{k_3}{k_4}P \rightarrow \log(Y) = \log\left(\frac{k_3}{k_4}\right) + \log(P)\end{aligned}$$

We want to find the extrinsic noise, the normalized covariance, which we have defined approximately by taking the covariance of the logarithm of the data:

$$\begin{aligned}n_{ext}^2 &= \text{Cov}(\log(X), \log(Y)) \\ &= \text{Cov}\left(\log\left(\frac{k_1}{k_2}\right) + \log(P), \log\left(\frac{k_3}{k_4}\right) + \log(P)\right) \\ &= \text{Cov}\left(\log\left(\frac{k_1}{k_2}\right), \log\left(\frac{k_3}{k_4}\right)\right) + \text{Cov}\left(\log\left(\frac{k_1}{k_2}\right), \log(P)\right) + \text{Cov}\left(\log(P), \log\left(\frac{k_3}{k_4}\right)\right) \\ &\quad + \text{Cov}(\log(P), \log(P))\end{aligned}$$

$$n_{ext}^2 = \text{Cov}(\log(P), \log(P)) = \text{Var}(\log(P))$$

To calculate intrinsic noise we need the total noise of each reporter:

$$\begin{aligned}n_{totX}^2 &= \text{Var}(\log(X)) = \text{Var}\left(\log\left(\frac{k_1}{k_2}\right) + \log(P)\right) = \text{Var}(\log(P)) \\ n_{totY}^2 &= \text{Var}(\log(Y)) = \text{Var}\left(\log\left(\frac{k_3}{k_4}\right) + \log(P)\right) = \text{Var}(\log(P))\end{aligned}$$

Which shows that in this example there is no intrinsic noise; hence a common promoter for two reporters is an extrinsic source of noise:

$$\begin{aligned}n_{intX}^2 &= n_{totX}^2 - n_{ext}^2 = 0 \\ n_{intY}^2 &= n_{totY}^2 - n_{ext}^2 = 0\end{aligned}$$

Suppose instead that there were two different plasmids with promoters P1 and P2 coding for reporters X and Y, and let their copy number be independent random variables. Setting up the problem the same way,

$$\begin{aligned}\frac{dX}{dt} &= k_1P_1 - k_2X \\ \frac{dY}{dt} &= k_3P_2 - k_4Y\end{aligned}$$

At steady-state,

$$X = \frac{k_1}{k_2} P_1 \rightarrow \log(X) = \log\left(\frac{k_1}{k_2}\right) + \log(P_1)$$

$$Y = \frac{k_3}{k_4} P_2 \rightarrow \log(Y) = \log\left(\frac{k_3}{k_4}\right) + \log(P_2)$$

Calculating the extrinsic noise,

$$\begin{aligned} n_{ext}^2 &= \text{Cov}(\log(X), \log(Y)) \\ &= \text{Cov}\left(\log\left(\frac{k_1}{k_2}\right) + \log(P_1), \log\left(\frac{k_3}{k_4}\right) + \log(P_2)\right) \\ &= \text{Cov}\left(\log\left(\frac{k_1}{k_2}\right), \log\left(\frac{k_3}{k_4}\right)\right) + \text{Cov}\left(\log\left(\frac{k_1}{k_2}\right), \log(P_2)\right) + \text{Cov}\left(\log(P_1), \log\left(\frac{k_3}{k_4}\right)\right) \\ &\quad + \text{Cov}(\log(P_1), \log(P_2)) \end{aligned}$$

$$n_{ext}^2 = \text{Cov}(\log(P_1), \log(P_2)) = 0$$

For the total noise,

$$\begin{aligned} n_{totX}^2 &= \text{Var}(\log(P_1)) \\ n_{totY}^2 &= \text{Var}(\log(P_2)) \\ n_{intX}^2 &= n_{totX}^2 - n_{ext}^2 = \text{Var}(\log(P_1)) \\ n_{intY}^2 &= n_{totY}^2 - n_{ext}^2 = \text{Var}(\log(P_2)) \end{aligned}$$

Hence in this case, where the random variable independently affects the two reporters, the extrinsic noise is zero, making these intrinsic noise sources.

Notice that the strength of our approach is when the two reporters are not identically regulated with identical statistics, as required by the Elowitz et al. derivation. We extend the applicability by assigning different extrinsic noise quantities to each reporter, so that now instead of there being a single extrinsic noise, each reporter has its own set of intrinsic and extrinsic contributions.

The following example shows what can happen to extrinsic noise in the case of negative feedback. Suppose we have the extrinsic promoter case, but reporter X has negative feedback (and $k_5 X \gg 1$):

$$\begin{aligned} \frac{dX}{dt} &= k_1 P \frac{1}{1 + k_5 X} - k_2 X \approx \frac{k_1 P}{k_5 X} - k_2 X \\ \frac{dY}{dt} &= k_3 P - k_4 Y \end{aligned}$$

At steady-state,

$$\begin{aligned} X^2 &= \frac{k_1 P}{k_2 k_5} \rightarrow \log(X^2) = \log\left(\frac{k_1}{k_2 k_5}\right) + \log(P) \rightarrow \log(X) = \frac{1}{2} \log\left(\frac{k_1}{k_2 k_5}\right) + \frac{1}{2} \log(P) \\ Y &= \frac{k_3}{k_4} P \rightarrow \log(Y) = \log\left(\frac{k_3}{k_4}\right) + \log(P) \end{aligned}$$

Calculating the extrinsic noise,

$$\begin{aligned}
 n_{ext}^2 &= \text{Cov}(\log(X), \log(Y)) \\
 &= \text{Cov}\left(k + \frac{1}{2}\log(P), \log\left(\frac{k_3}{k_4}\right) + \log(P)\right) \\
 &= \text{Cov}\left(k, \log\left(\frac{k_3}{k_4}\right)\right) + \text{Cov}(k, \log(P)) + \text{Cov}\left(\log(P), \log\left(\frac{k_3}{k_4}\right)\right) + \text{Cov}\left(\frac{1}{2}\log(P), \log(P)\right) \\
 n_{ext}^2 &= \text{Cov}\left(\frac{1}{2}\log(P), \log(P)\right) = \frac{1}{2}\text{Var}(\log(P))
 \end{aligned}$$

For the total noise,

$$\begin{aligned}
 n_{totX}^2 &= \text{Var}(\log(X)) = \text{Var}\left(k + \frac{1}{2}\log(P)\right) = \frac{1}{4}\text{Var}(\log(P)) \\
 n_{totY}^2 &= \text{Var}(\log(Y)) = \text{Var}\left(\log\left(\frac{k_3}{k_4}\right) + \log(P)\right) = \text{Var}(\log(P))
 \end{aligned}$$

If we calculate the intrinsic noise using the Elowitz et al. approach, we find that the extrinsic noise exceeds the total noise for reporter X. However, for this simplified example, we know the only noise source is an extrinsic variable, and thus the intrinsic noise should turn out to be zero. This allows us to infer that α , as described in the supplement, has a value of $\frac{1}{2}$, representing the fact that reporter X experiences half as much noise from the variable plasmid copy number as Y does.

$$\begin{aligned}
 n_{extX}^2 &= \frac{\alpha}{2}\text{Var}(\log(P)) = \frac{1}{4}\text{Var}(\log(P)) \\
 n_{extY}^2 &= \frac{1}{2\alpha}\text{Var}(\log(P)) = \text{Var}(\log(P)) \\
 n_{intX}^2 &= n_{intY}^2 = 0
 \end{aligned}$$

Recall that in the experiments, we must first estimate alpha in a case where the Elowitz et. al. assumptions hold, i.e., the reporters are identically regulated.

To further verify our analysis we use simulations to test the decomposition on noise for cases where we control the input intrinsic and extrinsic levels.

Figure 8: Simulations where the intrinsic (panel a) and extrinsic (panel b) noise change separately.

Illustrated in Figure 8a we first we vary the strength of transcription of a single bidirectional promoter coding for two fluorescent proteins, leading to perfectly correlated fluorescence quantities, which our decomposition shows to have only extrinsic noise and no intrinsic noise. Next, in Figure 8b we vary the strength of transcription of two fluorescent genes independently, which leads to uncorrelated fluorescence quantities; our method returns only intrinsic noise and no extrinsic noise.

Furthermore, we simulated mixtures of both noise types, generated by varying the strength of transcription of two different reporter genes as in the intrinsic noise case in the previous figure, but also by varying the amount of a transcription factor which regulates both genes. For Figure 9a, the noise breakdown gives an intrinsic noise of 0.22 and an extrinsic noise of 0.16 for both proteins. For Figure 9b, where we give the common transcription factor extra variability, it raises the extrinsic noise substantially (but not the intrinsic noise; the common transcription factor is an "extrinsic variable"). The intrinsic noise changes slightly to 0.21 and the extrinsic noise jumps to 0.23.

Figure 9: Simulations where the intrinsic and extrinsic noise change simultaneously.

Further comments:

2. The authors do not discuss possible molecular mechanisms behind the decomposed intrinsic and extrinsic noise. Do the authors aim at pure phenomenological noise decomposition (correlated and uncorrelated noise)?

The molecular mechanisms of intrinsic and extrinsic noise have been discussed extensively in literature. We avoided (deliberately) introducing extended discussion about the molecular mechanisms behind the noise contributions.

We cite several papers including the following:

- Elowitz, M., Levine, A., Siggia, E. & Swain, P. Stochastic Gene Expression in a Single Cell. *Science* 297, 1183-1186 (2002).
- Swain, P. S., Elowitz, M. B. & Siggia, E. D. Intrinsic and extrinsic contributions to stochasticity in gene expression. *Proceedings of the National Academy of Sciences U.S.A.* 99, 12795-12800 (2002).
- Pedraza, J. & van Oudenaarden, A. Noise Propagation in Gene Networks. *Science* 307, 1965-1969 (2005).
- Mads Kærn, Timothy C. Elston, William J. Blake & James J. Collins. Stochasticity in gene expression: from theories to phenotypes. *Nature Reviews Genetics* 6, 451-464 (June 2005).
- Jonathan M. Raser, and Erin K. O'Shea. Control of Stochasticity in Eukaryotic Gene Expression. *Science* 304 (5678): 1811-1814 (2004)

As the reviewer suggests, we view this decomposition as the result of processing the experimentally observed response of the circuits under the following formulation: Let Y be the constitutive reporter and X a regulated reporter. A is a function of all extrinsic variables, B and C are the intrinsic variables for each promoter (independently of the sources of noise), and α is a coefficient which is 1 for two constitutive promoters with identical reporter statistics but varies depending on the regulation of X . In other words, α represents an aggregated susceptibility to fluctuating variables which affect both reporters: $X = A^\alpha B$, $Y = AC$.

3. Some of the terminology is confusing. Why are the synthesis and degradation of proteins named thermal events (supplementary information)? Which processes in a cell are thermal events and which are not?

This wording was meant to distinguish intrinsic noise, as opposed to extrinsic noise. The terminology has been updated and the wording “thermal events” is removed from the supplement.

4. The authors do not specify if the constructs are chromosomally integrated. If yes, they should verify where the integration events occurred. It would be useful to show results from multiple different chromosomal integrations. This would certainly help to illustrate the decomposition of noise associated with the different chromosomal localizations (see. Eg. Becskei et al (2005, Nature Genetics), Batenchuk et al (2011, Biophysical J)).

Indeed the circuits are stably integrated in the cells. As the title suggests we are working with transgenes and in the methods section we discuss the generation of stable cell lines. To stably introduce the DNA in cells we transfected the plasmid carrying the circuit with a Hygromycin linear selection marker. Cells were transferred to petri dishes and incubated in Hygromycin. Colonies positive for florescent were selected by microscopy and picked using cloning rings and further expanded.

We did not proceed to locate the integration site for two reasons. First, we argue that our ability to control the transcription levels of the transgene emulates the effect of variable transcriptional activity in the promoter (which is to be expected for different integration sites). Specifically, Figure 3 of the response document (and Supplement Figure 3) corresponds to the coefficient of variation for different strength of transcription as controlled by the levels of doxycycline. The behavior of the negative feedback is consistent for different expression levels. Secondly, we argue that simply knowing the actual location would not allow us to make any general statements about the behavior of the transgenes. To experimentally probe the effect of the location on the expression levels and noise we created new integrations of the negative feedback architecture and we performed titrations of IPTG for high doxycycline levels. As illustrated in the response document Figure 1, a direct comparison between the three clones shows that the behavior between the transgenes is highly consistent. Furthermore, we performed the noise decomposition and the results (Figure 10) show consistent behavior. These results we included as Supplement Figure 11.

Figure 10: Noise decomposition for the negative feedback transgenes.

5. The authors should provide the value of background fluorescence of cells not containing any GFP and RFP in Fig. 1.

Given the restructuring of the paper and the additional comments on the microscopy snapshots we removed these images from the main paper Figure 1. In any case, the background fluorescence was zsGreen1: 1640 and dsRed:1800 for the negative feedback snapshots and zsGreen1: 2500 and dsRed:2400 for the simple regulation snapshots. For the negative feedback,

the exposure times were ZsGreen1:500ms and dsRed:1000ms, while for the simple regulation, the exposure times were ZsGreen1:400ms and dsRed:2000ms

6. It is quite unusual that the total noise reduces with increasing mean expression (Fig. 4a, c). Do the authors have an explanation for this?

We assume that the reviewer suggests that it is unusual to see the total noise reduce with decreasing mean expression. The role of negative feedback in cells is a matter of continued debate. Reviewer 1 mentioned that he was not overly surprised by the results.

We cite several papers that demonstrate the possible increase or decrease of noise by negative autoregulation (Introduction, first paragraph). As an example from the literature, if we define the operator and repressor dissociation kinetic as k_{ar} and the mRNA synthesis rate as α_0 , Cox et al¹⁵ show that the mean is dependent on the ratio $\kappa_1 = k_{ar}/\alpha_0$. For $\kappa_1 > 1$ (fast operator dynamics), it was shown that an inverse scaling of the mean and CV^2 exists whereas for $\kappa_1 < 1$ (slow operator dynamics), a non-inverse scaling of the mean and the CV^2 exists. Hence both kinds of scaling of mean and the CV^2 are theoretically possible. To conclude, we believe that the above theoretical results warrant additional experimental investigation, particularly in human cells.

7. More details should be provided for the experimental noise decomposition. Each step (15-20) should be presented in the supplementary information. For example, what the value for alpha is.

We significantly revised both the paper and the supplement on the noise decomposition. We introduced a new section in the supplement with a motivating example. We also provide the values of alpha for the experiments in the manuscript. The following table is included in the supplement.

IPTG (μM)	Negative Feedback High Dox	Negative Feedback Low Dox	Simple Negative Regulation High Dox	Simple Negative Regulation Low Dox
50	1	1.22	1.11	1.28
25	1.046293	1.280207	1.170337	1.332335
12.5	0.986329	1.211581	1.232468	1.230991
6.25	0.961434	1.195583	1.064725	1.084425
3.125	0.847654	1.111019	0.868676	0.861535
1.6	0.692237	0.864742	0.684977	0.626666
0.8	0.64337	0.749677	0.548974	0.431871
0.4	0.605457	0.677382	0.389802	0.303799
0.2	0.568188	0.649937	0.323275	0.265747
0	0.5828	0.630114	0.337724	0.23634

8. It would be very helpful to plot the noise terms against the mean expression level and not only against IPTG.

We plotted the noise against the mean levels and the following panels are included in the main paper Figure 2 and response document Figure 3.

References

1. Newman, J. R. S. *et al.* Single-cell proteomic analysis of *S. cerevisiae* reveals the architecture of biological noise. *Nature* **441**, 840-846 (2006).
2. Singh, A., Razooky, B., Cox, C. D., Simpson, M. L. & Weinberger, L. S. Transcriptional Bursting from the HIV-1 Promoter Is a Significant Source of Stochastic Noise in HIV-1 Gene Expression. *Biophys. J.* **98**, L32-L34 (2010).
3. Simpson, M. L., Cox, C. D. & Saylor, G. S. Frequency domain analysis of noise in autoregulated gene circuits. *Proc Natl Acad Sci U S A* **100**, 4551-4556 (2003).
4. Austin, D. W. *et al.* Gene network shaping of inherent noise spectra. *Nature* **439**, 608-611 (2006).
5. Rosenfeld, N., Young, J., Alon, U., Swain, P. & Elowitz, M. Gene regulation at the single-cell level. *Science* **307**, 1962-5 (2005).
6. Bar-Even, A. *et al.* Noise in protein expression scales with natural protein abundance. *Nature Genetics* **38**, 636-643 (2006).
7. Swain, P. S., Elowitz, M. B. & Siggia, E. D. Intrinsic and extrinsic contributions to stochasticity in gene expression. *Proceedings of the National Academy of Sciences* **99**, 12795-12800 (2002).
8. Volfson, D. *et al.* Origins of extrinsic variability in eukaryotic gene expression. *Nature* **439**, 861-864 (2006).
9. Hoebeek, J., Speleman, F. & Vandesompele, J. in (eds Hilario, E. & Mackay, J.) 205-226 (Humana Press, 2007).
10. Bodin, L., Beaune, P. H. & Loriot, M. Determination of Cytochrome P450 2D6 (CYP2D6) Gene Copy Number by Real-Time Quantitative PCR. *Journal of Biomedicine and Biotechnology* **2005**, 248-253 (2005).
11. Zheng, S. *et al.* DNA hypermethylation profiles associated with glioma subtypes and EZH2 and IGFBP2 mRNA expression. *Neuro-oncology* **13**, 280-289 (2011).
12. Li, Y., Moore, R., Guinn, M. & Bleris, L. Transcription activator-like effector hybrids for conditional control and rewiring of chromosomal transgene expression. *Scientific Reports* **2:897** (2012).
13. Alon, U. in *An Introduction to Systems Biology: Design Principles of Biological Circuits (Chapman & Hall/Crc Mathematical and Computational Biology Series)* ({Chapman & Hall/CRC}, 2006).

14. Bleris, L. *et al.* Synthetic incoherent feedforward circuits show adaptation to the amount of their genetic template. *Mol Syst Biol* **7**, 519 (2011).
15. Cox, C. D., McCollum, J. M., Allen, M. S., Dar, R. D. & Simpson, M. L. Using noise to probe and characterize gene circuits. *Proceedings of the National Academy of Sciences* **105**, 10809-10814 (2008).

Thank you again for submitting your work to Molecular Systems Biology. We have now heard back from the two referees who accepted to evaluate the revised study. As you will see, the referees are now supportive. They have however some remaining points, which we would ask you to address with appropriate amendments in the text.

Please resubmit your revised manuscript online, with a covering letter listing amendments and responses to each point raised by the referees. Please resubmit the paper ****within one month**** and ideally as soon as possible. If we do not receive the revised manuscript within this time period, the file might be closed and any subsequent resubmission would be treated as a new manuscript. Please use the Manuscript Number (above) in all correspondence.

Click on the link below to submit your revised paper.

<<http://mts-msb.nature.com/cgi-bin/main.plex?el=A1BL5Byb7B4FH5I4A9zPqJOnJ8pC0oHK5su1MejAZ>>

If you do choose to resubmit, please use the link below to access the Licence to Publish. Please complete and sign this on behalf of all authors, with their consent, and fax to +44 (0)1256 321670.

<http://www.nature.com/licenceforms/msb/msb-ltp-cc-by-sa-nd.pdf>

Processing of your submission can proceed when we have received this form.

REFeree REPORTS:

Reviewer #2 (Remarks to the Author):

I was not fully satisfied with the response provided by the authors. First, at a somewhat technical level, the description of changes to the text is very incomplete: There are no precise pointers to all changes made in the text; and in the manuscript itself, the revised text is not highlighted in any way. Thus, I am unable to directly identify how the text has changed in response to each of my (as well as other reviewers') comments. As I'm sure the editor knows, listing all changes and highlighting them in the text are standard procedure in "response to reviewers" documents. The MSB editorial team should have insisted on this standard.

Within the existing response letter, I did not find a satisfying response to some of my main concerns. In particular:

1) In my review, I explicitly stated that "my main concern is that there is no feedback between the theoretical and experimental components", and asked to see a direct comparison between theoretical predictions and experimental measurements. As an alternative, I suggested the use of numerical simulations as an intermediate between analytical theory and experiment. In their response, the authors thank me for "insightful and constructive comments" but do not describe any action taken in response to my comments.

2) In my follow up comment, I challenged the authors' assertion that negative feedback reduces gene expression noise. In response, the authors do re-plot some of their data in a format that I requested, but again they do not address the essence of my question itself: Do they stand behind their original statement? Does their data support it? If so, what is their take on past work claiming the contrary?

Reviewer #3 (Remarks to the Author):

The authors made changes that significantly improved the presentation of the paper. They also performed experiments to address how chromosomal position affects the behavior of negative feedback. Upon the following minor changes, I suggest the paper for publication.

Q1 & Q7. The inclusion of several examples (two constitutive promoters on identical and distinct plasmids; negative feedback) helped to clarify the nature of noise decomposition. The molecular basis of $\alpha = 1/2$ is explained. While it is stated that experimentally obtained values for α are inserted in the manuscript, I could not find them. In case I have not overlooked them, they should be provided.

Q4. The determination of chromosomal position does help shed light on molecular mechanisms but forms the basis of comparison with future experimental works aimed at elucidating chromosomal position effect. Since this identification has not been performed by the authors, they should at least state that the three colonies were selected randomly for further investigation.

Q6. The fact that noise is proportional with the mean expression is hypothesized to be due to slow operator dynamics. Since in mammalian cells, the fluctuations are not necessarily explained by the operator only (but e.g. initiation,...), I suggest using a more general term.

Point-by-point response to reviewers' comments

We thank again the reviewers for their comments and remarks. We hope that the reviewers will find our answers satisfactory. Please note that the original comments are in blue color.

For all the co-authors,
Leonidas Bleris

Reviewer #2 (Remarks to the Author):

Review: Shimoga et al. (2nd round)

I was not fully satisfied with the response provided by the authors. First, at a somewhat technical level, the description of changes to the text is very incomplete: There are no precise pointers to all changes made in the text; and in the manuscript itself, the revised text is not highlighted in any way. Thus, I am unable to directly identify how the text has changed in response to each of my (as well as other reviewers') comments. As I'm sure the editor knows, listing all changes and highlighting them in the text are standard procedure in "response to reviewers" documents. The MSB editorial team should have insisted on this standard.

The paper was restructured to a short format; several paragraphs were removed and new were added. This hindered our ability to directly highlight changes, yet we did mention the location in our responses. We apologize for this technical issue. To assist the reviewer, we list the exact location for each of the major questions:

Reviewer 1

Major Q1: Paper page 6/paragraph 3, Paper page 9/paragraph 3, Supplement Figure 11.

Major Q2: Paper page 9/paragraph 2, Supplement Figure 10.

Major Q3: Paper page 10/paragraph 3.

Reviewer 2

Major Q1: Paper Figure 3.

Major Q2: We introduced the requested citations, Paper page 3/paragraph 4.

Major Q3: Paper page 6/paragraph 3, Methods and Supplementary Material Integration Copies, Supplement Figure 11, Supplementary Material Table I.

Major Q4: The paragraph was restructured, Paper page 5/paragraph 1.

Major Q5: Supplementary Material Theory, starting "Such a multiplicative model can be motivated as follow..."

Major Q6: We added the scale bars.

Reviewer 3

Major Q1: Supplementary material Verification and decomposition of simulated noise, Supplementary Figure 13 and Figure 14.

Within the existing response letter, I did not find a satisfying response to some of my main concerns. In particular:

1) In my review, I explicitly stated that "my main concern is that there is no feedback between the theoretical and experimental components", and asked to see a direct comparison between theoretical predictions and experimental measurements. As an alternative, I suggested the use of numerical simulations as an intermediate between analytical theory and experiment. In their response, the authors thank me for "insightful and constructive comments" but do not describe any action taken in response to my comments.

We apologize for this issue. We should have included discussion to direct your attention to our response to Reviewer 3, Question 1. In particular, to clarify and further verify our analysis we used simulations to test the decomposition of noise for cases where we control the input intrinsic and extrinsic levels. Using this approach we perform a direct comparison between numerical simulations and our analytical theory. These results are included in the Supplementary material, Verification and decomposition of simulated noise, Supplementary Figure 13 and Figure 14. Furthermore, we added a discussion in the Paper page 9/paragraph 2, starting with “We also used simulations to gain additional insight...”

We did not perform nor do we have anything new to offer in terms of explicitly modeling and simulating negative feedback. There are several modeling papers which we cite in the manuscript. With respect to the intrinsic and extrinsic decomposition of simulated data, a number of models (e.g. (1, 2)) predict that the intrinsic noise goes up with stronger repression which is consistent with our conclusions. Additionally, other models predict (e.g. (3, 4)) that extrinsic noise goes down with stronger repression which is also consistent with our conclusions.

2) In my follow up comment, I challenged the authors' assertion that negative feedback reduces gene expression noise. In response, the authors do re-plot some of their data in a format that I requested, but again they do not address the essence of my question itself: Do they stand behind their original statement? Does their data support it? If so, what is their take on past work claiming the contrary?

Using our assumptions and theoretical formulation for the analysis of the experimental data we show that the extrinsic noise goes down and the intrinsic noise goes up with stronger repression. Importantly, the total noise goes down with stronger repression. The first two conclusions are derived using our analysis and processing of the data and the last conclusion can be derived directly from the flow cytometry distributions. We stand behind our original assessment that negative feedback can decrease total noise in mammalian synthetic transgenes and our data indeed support that statement.

As we mentioned previously, a number of models are consistent with our conclusions, and predict that the intrinsic noise goes up and that extrinsic noise goes down with stronger repression. Nonetheless, we wish to emphasize that these models are approximate, and in fact a detailed computational study shows that, for certain parameter regimes, it could be the case that intrinsic noise can also go down with stronger repression (5). We expect that the role of negative feedback in cells will remain a matter of continued debate.

With respect to “past work that claims the contrary”, indeed theoretical results show that this architecture might either amplify or reduce the noise in gene expression (5-7), highlighting the need for further experimental investigation. To date, the vast majority of experiments have been performed in bacteria and yeast (4, 8, 9), and to our knowledge this is the first study of a synthetic negative feedback architecture stably integrated in human cells.

Reviewer #3 (Remarks to the Author):

The authors made changes that significantly improved the presentation of the paper. They also performed experiments to address how chromosomal position affects the behavior of negative feedback. Upon the following minor changes, I suggest the paper for publication.

Thank you for your comments and support.

Q1 & Q7. The inclusion of several examples (two constitutive promoters on identical and distinct plasmids; negative feedback) helped to clarify the nature of noise decomposition. The molecular basis of $\alpha = 1/2$ is explained. While it is stated that experimentally obtained values for alpha are inserted in the manuscript, I could not find them. In case I have not overlooked them, they should be provided.

The obtained values for alpha are included in the supplementary material as Table II.

Q4. The determination of chromosomal position does help shed light on molecular mechanisms but forms the basis of comparison with future experimental works aimed at elucidating chromosomal position effect. Since this identification has not been performed by the authors, they should at least state that the three colonies were selected randomly for further investigation.

Thank you for the remark, we introduced the following edit: “We created new integrations for the negative feedback, selected three random colonies and we performed titrations of IPTG for high doxycycline levels.”

Q6. The fact that noise is proportional with the mean expression is hypothesized to be due to slow operator dynamics. Since in mammalian cells, the fluctuations are not necessarily explained by the operator only (but e.g. initiation,...), I suggest using a more general term.

We do not generally attribute the observation to the slow operator dynamics, rather we included a specific example from the literature (10) to simply illustrate that both kinds of scaling of mean are theoretically possible. Additional papers (main paper introduction) argue the same. We note that the Cox et al paper includes additional discussion specific to eukaryotes.

References

- 1. Stekel D, Jenkins D. Strong negative self-regulation of prokaryotic transcription factors increases the intrinsic noise of protein expression. *BMC Systems Biology*. 2008; 2(1): 6.**
- 2. Thattai M, van Oudenaarden A. Intrinsic noise in gene regulatory networks. *Proc Natl Acad Sci USA*. 2001; 98: 8614-8619.**

3. Austin DW, Allen MS, McCollum JM, Dar RD, Wilgus JR, Sayler GS, Samatova NF, Cox CD, Simpson ML. Gene network shaping of inherent noise spectra. *Nature*. 2006 02/02; 439(7076): 608-611.
4. Dublanche Y, Michalodimitrakis K, Kummerer N, Foglierini M, Serrano L. Noise in transcription negative feedback loops: Simulation and experimental analysis. *Mol Syst Biol*. 2006; 2.
5. Marquez-Lago TT, Stelling J. Counter-intuitive stochastic behavior of simple gene circuits with negative feedback. *Biophys J*. 2010 5/5; 98(9): 1742-1750.
6. Singh A, Hespanha JP. Optimal feedback strength for noise suppression in autoregulatory gene networks. *Biophys J*. 2009 05/20; 96(10): 4013-4023.
7. Thattai M, van Oudenaarden A. Intrinsic noise in gene regulatory networks. *Proc. Natl. Acad. Sci. U.S.A.* 2001; 98(15): 8614-8619.
8. Becskei A, Serrano L. Engineering stability in gene networks by autoregulation. *Nature*. 2000; 405: 590-593.
9. Nevozhay D, Adams RM, Murphy KF, Josić K, Balázsi G. Negative autoregulation linearizes the dose-response and suppresses the heterogeneity of gene expression. *Proceedings of the National Academy of Sciences*. 2009 March 31; 106(13): 5123-5128.
10. Cox CD, McCollum JM, Allen MS, Dar RD, Simpson ML. Using noise to probe and characterize gene circuits. *Proceedings of the National Academy of Sciences*. 2008 August 05; 105(31): 10809-10814.